# Recent Developments in Reactions and Catalysis of Protic Pyrazole Complexes

**DOI:** 10.3390/molecules28083529

**Published:** 2023-04-17

**Authors:** Wei-Syuan Lin, Shigeki Kuwata

**Affiliations:** 1Department of Chemical Science and Engineering, School of Materials and Chemical Technology, Tokyo Institute of Technology, 2-12-1 E4-1 O-okayama, Meguro-ku, Tokyo 152-8552, Japan; lin.w.ae@m.titech.ac.jp; 2Department of Applied Chemistry, College of Life Sciences, Ritsumeikan University, 1-1-1 Noji-higashi, Kusatsu 525-8577, Shiga, Japan

**Keywords:** pyrazole, pyrazolato, pincer ligand, metal–ligand cooperation, homogeneous catalysis, bifunctional catalysis, transfer hydrogenation, hydrazine

## Abstract

Protic pyrazoles (*N*-unsubstituted pyrazoles) have been versatile ligands in various fields, such as materials chemistry and homogeneous catalysis, owing to their proton-responsive nature. This review provides an overview of the reactivities of protic pyrazole complexes. The coordination chemistry of pincer-type 2,6-bis(1*H*-pyrazol-3-yl)pyridines is first surveyed as a class of compounds for which significant advances have made in the last decade. The stoichiometric reactivities of protic pyrazole complexes with inorganic nitrogenous compounds are then described, which possibly relates to the inorganic nitrogen cycle in nature. The last part of this article is devoted to outlining the catalytic application of protic pyrazole complexes, emphasizing the mechanistic aspect. The role of the NH group in the protic pyrazole ligand and resulting metal–ligand cooperation in these transformations are discussed.

## 1. Introduction

Pyrazole is an aromatic five-membered N-heterocycle containing a potentially Brønsted acidic NH group adjacent to a Schiff-base nitrogen atom. This amphiprotic character gives rise to the rich coordination chemistry of pyrazoles. Unlike aprotic N-heterocycles, such as pyridine, pyrazole can be deprotonated easily, and the resulting pyrazolate anion bridges two metal centers to form di- or polynuclear complexes in some cases. The NH group also provides a clue to integration of pyrazole units, making multidentate ligands, such as poly(pyrazolyl)borates [1]. The flexible ligand design based on the easy construction of the pyrazole ring [2,3,4] and N-functionalization has led to the structural diversity of the pyrazole complexes [5,6,7,8] and their applications in various fields, including materials chemistry [9], homogeneous catalysis [10,11], bioinorganic modeling [12], supramolecular chemistry [13,14,15], and medicinal chemistry [16].

Coordination of a pyrazole to a Lewis acidic metal center renders the pyrazole NH proton more acidic [2]. The increased Brønsted acidity causes intra- and intermolecular hydrogen bonding as well as facile deprotonation that switches the coordination mode from a lone-pair donating L-type to a covalent X-type. These events should make an electronic impact on the complex. It is to be emphasized that the deprotonation in the position β to the metal may be coupled with ligand dissociation on the metal center. As illustrated in Figure 1a, elimination of HX or outer sphere transfer of nucleophilic group X along with a proton from such a “β-protic” pyrazole complex would yield a coordinatively unsaturated pyrazolato complex. Bond cleavage of pronucleophile HX on this complex regenerates the pyrazole complex. The interconversion associated with change in the coordination mode of the pyrazole rather than the formal oxidation state of the metal would mediate various bond activation and transfer of HX. Such metal–ligand cooperative transformations have also been known for related proton-responsive ligands [17,18,19,20,21,22,23,24,25,26], exemplified in Figure 1b. Nevertheless, difference in ligand acidity and relative positions to the metal center in the protic pyrazole complexes brings about their unique reactivities. Tamm and co-workers demonstrated that the pyrazolylborane **1** reacts with dihydrogen gas at room temperature to afford the zwitterionic pyrazolium borate **2** (Figure 2) [27]. Heterocumulenes, such as carbon dioxide, also react with **1** to give the corresponding adduct **3**, for example [28]. These reactions showcase the pyrazole-based bifunctional reactivities operating even in the field of transition-metal-free, frustrated Lewis pair chemistry.

In this review, we outline the reactivities of the β-protic pyrazole complexes of transition metals. The proton-responsive nature and catalytic application of this class of complexes were surveyed about a decade ago [10,11]. This article updates these earlier reviews by focusing on their stoichiometric and catalytic reactivities. Special attention will be paid to the mechanistic aspects in the discussion on the catalysis. Protic pyrazoles are also known as modules in spin-crossover materials [29,30,31,32,33] and phosphorescent complexes [34], which are not covered in this review. The readers can also refer to papers [33,35,36] to learn the design and synthesis of specific classes of protic pyrazole ligands.

## 2. Reactions of Protic Pyrazole Complexes

### 2.1. Pincer-Type Complexes Bearing Protic Pyrazole Arms

Chelation has been a rational strategy to ensure the coordination of pyrazoles for metal–ligand cooperative reactivities [35,37]. During the last decade, significant advances have been made particularly in the coordination chemistry of 2,6-bis(1*H*-pyrazol-3-yl)pyridines (^R^LH_2_; R represents the substituent at the 5-position of the pyrazole ring), which place the two protic pyrazole groups at trans positions rigorously, owing to the rigid pincer-type framework [38]. This class of compounds have long been used as a ligand in iron(II) complexes showing thermal and photochemical spin-state transitions [29,30,31,32,33]. The protic pyrazole groups therein greatly affect the spin-crossover properties of the complexes through hydrogen bonding interaction with surrounding counteranion and solvent. In this section, we describe the transition metal ^R^LH_2_ complexes and related pincer-type pyrazole complexes, focusing on their stoichiometric reactivities, which originate mostly from the protic pyrazole units.

#### 2.1.1. Bis(1*H*-pyrazol-3-yl)pyridine Complexes

##### Ruthenium and Osmium

In 2010, Thiel and co-workers reported that deprotonation of the *^n^*^Bu^LH_2_-ligated ruthenium(II) complex **4a** under carbon monoxide leads to the formation of the bis(pyrazolato) carbonyl complex **5** (Figure 3) [39]. The result shows the diprotic nature of the pincer-type complex **4a**. The *^n^*^Bu^L complex **5** is also obtained by the dehydrogenative coordination of *^n^*^Bu^LH_2_ to [RuH_2_(CO)(PPh_3_)_3_]. 

Soon after that, our group demonstrated that the deprotonation can be done stepwise and reversible at least in the first step [40]. Thus, the ruthenium(II) complex **4b** having a *tert*-butyl substituted pincer ligand, *^t^*^Bu^LH_2_, undergoes reversible deprotonation by equimolar amount of a base to afford the pyrazole–pyrazolato complex **6** (Figure 4). The partial deprotonation of the pincer ligand is established by the ^1^H NMR spectrum of **6**, showing the D_2_O-exchangeable NH resonance at δ 10.17 with only 1H intensity as well as inequivalence of the two pyrazole arms. The single crystal X-ray analysis allows the detailed structural comparison between the pyrazole and deprotonated pyrazolato rings. The deprotonated pyrazole group lacks neighboring hydrogen bond acceptors and features a smaller N_α_N_β_C angle (106.8(4)°) due to the increased s character of the lone pair electrons on the deprotonated nitrogen atom [10,41]. Complete deprotonation of the pyrazole arms is achieved by an additional base in methanol, giving the bis(pyrazolato) methanol complex **7**. In the crystal of **7**, a hydrogen bonding network involving a co-crystalized methanol is observed (inset). The methanol ligand in **7** is replaced by molecular nitrogen and oxygen to yield the dinitrogen complex **8** and a side-on peroxo complex, respectively.

Some osmium complexes having ^R^LH_2_ or its deprotonated form are known [42,43]. Substitution reactions of [OsCl_2_(H_2_O)(*^t^*^Bu^LH_2_)] by bio-relevant molecules are reported [43].

##### Rhodium and Iridium

Goldberg and co-workers reported the reactivities of the iridium(I) complex [Ir(*^t^*^Bu^LH_2_)(coe)_2_]PF_6_ (**9**; coe = cyclooctene) bearing labile coe ligands (Figure 5) [44]. The coe ligands are easily displaced by carbon monoxide, giving the bis(carbonyl) complex **11** featuring *κ*^2^-coordination of the *^t^*^Bu^LH_2_ ligand. When **11** is dissolved in acetone-*d*_6_ under N_2_, a monocarbonyl species **10** having equivalent pyrazole arms is observed, suggesting an equilibrium between **10** and **11**. Addition of 4-*tert*-butylpyridine (*t*BuPy) to **11** results in an oxidative addition of one of the two pyrazole NH groups to afford the hydridoiridium(III) pyrazolato complex **12**. Co-crystallization of an additional pyridine, which is engaged in hydrogen bonding with the pyrazole arm of the *^t^*^Bu^LH ligand, has been confirmed by X-ray analysis. Deprotonation of **12** affords the bis(pyrazolato) complex [IrH(CO)(*t*BuPy)(*^t^*^Bu^L)]. Triphenylphosphine also reacts with the carbonyl complex **11** with oxidative addition of the pyrazole arm; however, the isolated product is [IrH(CO)(PPh_3_)_2_(*^t^*^Bu^LH)]PF_6_, having a *κ*^2^-bound *^t^*^Bu^LH ligand. Interestingly, thermolysis of **12** leads to dinuclear reductive elimination of H_2_ to give the diiridium(II) complex **13**. Complex **13** is also formed by treatment of **12** with (Ph_3_C)PF_6_. Deprotonation of the dicationic complex **13** yields the bis(pyrazolato) complex [{Ir(CO)(*t*BuPy)(*^t^*^Bu^L)}_2_]. Isolation of the iridium complexes ranging from Ir(I) to Ir(III) indicates the electronic flexibility of the ^R^LH*_n_* (*n* = 0–2) ligands.

On the other hand, Bogojeski and Bugarčić [45] and we [46] reported the synthesis and crystal structures of the trichlorido rhodium(III) and iridium(III) complexes [MCl_3_(*^t^*^Bu^LH_2_)] (M = Rh, Ir), respectively. The kinetics of the substitution reactions of the rhodium complex with small biomolecules, such as amino acids, is studied [45].

##### Platinum

Goldberg and co-workers [47] demonstrated that reaction of the chlorido complex **14** with methyllithium results in methylation of the platinum center, along with deprotonation of the *^t^*^Bu^LH_2_ ligand to afford the methyl complex **15** with a dangling lithium cation on one of the two pyrazolato arms (Figure 6). In the solution of THF-*d*_8_, the pincer ligand in **15** is symmetric, and the lithium cation is removed by the treatment with bis(triphenylphosphine)iminium (PPN) chloride to give the anionic complex **16**. The methyl bis(pyrazolato) complex **16** undergoes three-step protonation with a pyridinium tetrafluoroborate. While the first protonation product is insoluble and remains uncharacterized, following protonation yields the cationic methyl complex **17** and acetonitrile complex **18** sequentially. Thus, the first site of protonation is the pyrazolato ligand rather than the Pt–Me bond. Methane release from [PtMe(*^t^*^Bu^LH_2_)]Cl, a chloride salt of **17**, is not observed in C_6_D_6_ until 180 °C, indicating that the high energy barrier for the intramolecular proton transfer to the methyl ligand.

In this connection, uncharged bis(pyrazolato) phosphine complexes [Pt(PPh_3_)(^R^L)] are also synthesized [48]. The complexes display green phosphorescence in solution and in the solid state; however, the reactivities are unknown.

##### First-Row Transition Metals

The 3d transition metal complexes of ^R^LH_2_ have been known much earlier [49]. Most of the studies were, however, limited to their structural determination, except for the spin-crossover properties of the iron(II) complexes with the Fe:ligand ratio of 1:2 until recently. In 2013, we synthesized a 1:1 complex **19** of iron(II) and *^t^*^Bu^LH_2_ and uncovered the reactivity [50]. The paramagnetic dichlorido complex **19** is converted to the diamagnetic phosphine complex **20a** with the aid of sodium triflate. Complex **20a** catalyzes disproportionation of hydrazine (vide infra). We later obtained the cobalt and manganese analogues [MCl_2_(*^t^*^Bu^LH_2_)] (M = Co (**21b**), Mn) and explored the ligand substitution reactions of **19** and **21b**, as summarized in Figure 7 [49]. The iron and cobalt complexes, **19** and **21b**, are converted into the high-spin, triflato complexes **22**, upon treatment with silver triflate in acetonitrile. The iron complex **22a** reacts with dioxygen to give the oxido-bridged diiron(III) complex **23** [49,51]. The two pincer ligands in **23** are almost perpendicular, and the protic pyrazole arms make a hydrogen bond with the triflato ligand on the opposite iron center. The low-spin phosphine and carbonyl complexes, **20** and **24**, can further be derivatized by sequential ligand replacement of the triflato complexes of **22**. The ammine complex [Fe(NH_3_)(PMe_3_)_2_(*^t^*^Bu^LH_2_)](OTf)_2_ is obtained similarly [50].

Following that, Caulton and co-workers reported dehydrochlorination of the iron(II) and cobalt(II) complexes **21** with two equiv of a lithium silylamide in THF (Figure 8) [52,53]. Full dehydrochlorination is, however, not achieved, and the products are the monochlorido-bridged, anionic complexes **25**. For cobalt, the reactions in diethyl ether or toluene yield single crystals of unexpected polynuclear complexes, apparently caused by partial dissociation of the *^t^*^Bu^LH_2_. Complexes **25** can be regarded as lithium chloride adducts of dimers of the expected two-fold dehydrochlorination products M*^t^*^Bu^L with coordinative unsaturation.

In contrast, treatment of the iron complex **21a** in the presence of two-electron donor ligands results in complete dissociation of the chloride ligands to give the bis(pyrazolato) complexes, such as **26**–**28** (Figure 9) [52]. The 4-dimethylaminopyridine (DMAP) complex **26** is paramagnetic, while the isocyanide complex **27** as well as the diphosphine complex **28** obtained similarly is diamagnetic. The DMAP complex **26** is further converted to the oxido-bridged Fe(III)_2_ complex **29** upon treatment with silver triflate, although the oxidant remains unclear [54]. In **29**, the Lewis-basic pyrazolate arms are bridged by the silver cation. On the other hand, chloride abstraction of **21a** with two equiv of NaBAr^F^_4_ (Ar^F^ = C_6_H_3_(CF_3_)_2_-3,5) in THF leads to the formation of the dicationic THF complex **30** [52], which, however, is found to decompose into [Fe(*^t^*^Bu^LH_2_)_2_]^2+^ with ligand redistribution during recrystallization from dichloromethane.

As in the case of the iron analogue **21a**, dehydrochlorination of the cobalt(II) complex **21b** in the presence of triethylphosphine affords the bis(phosphine) complex **31** (Figure 10) [55]. Subsequent treatment with nitrous oxide results in oxidation of the phosphine ligand to yield the (phosphine oxide)-bridged dinuclear complex **32**. This complex catalyzes oxidation of PEt_3_ and PPh_3_ with nitrous oxide or O_2_, owing to the interconversion between **31** and **32**. Meanwhile, the ligand basicity of **31** was suggested by the formation of the cobalt(III) complex **33** with each pyrazolato arm binding to a silver center [54]. Even cooperation of Lewis acid–Brønsted base centers is proposed for the oxidation of **31** in dichloromethane to give the cobalt(III) chlorido complex **34** (Figure 11) [56].

Synthesis of the elusive, coligand-free bis(pyrazolato) complexes M^R^L has been achieved for chromium(II) by redox-neutral, salt metathesis reaction of K_2_*^t^*^Bu^L [54] with CrCl_2_ or treatment of a chromium(II) silylamide and *^t^*^Bu^LH_2_ [57]. The primary product was claimed to be the bis(pyrazolato)-type dimer **35**, which subsequently converts into the tetramer **36** during recrystallization (Figure 12). Owing to the coordinative unsaturation, **35** reacts with four equiv of DMAP to give the mononuclear bis(dmap) complex **37** with a square-pyramidal geometry. In contrast, addition of chloride anion to **35** affords the anionic monochlorido complex **38**, maintaining the pyrazolato-bridged dichromium(II) core.

Cooperative reactivities of the Lewis acidic metal center and Brønsted basic pyrazolato nitrogen atoms in **35** are also reported. Treatment of **35** with phenol does not result in simple redox-neutral acid–base reaction to give a phenoxido–pyrazole complex; instead, the Cr(II)Cr(III) mixed-valent phenoxido complex **39a** is obtained (Figure 12) [57]. The oxidation has been explained by evolution of dihydrogen gas, which, however, is not detected in the reaction mixture. Benzoic acid similarly reacts with **35** to yield the corresponding benzoate complex **39b**. On the other hand, treatment of **35** with an equimolar amount of 2-naphthol (NapOH) results in partial dissociation of the pincer ligand to afford the trinuclear Cr(II)Cr(III)_2_ complex [{(*^t^*^Bu^LH_2_)Cr(ONap)(µ_2_-ONap)_2_}_2_Cr](Cl)_2_, wherein the two chloride counteranions are accommodated between the pyrazole rings through hydrogen bonding [57]. 

Chemical reduction of **35** with four equiv of KC_8_ suggests that the pincer-type bis(1*H*-pyrazol-3-yl)pyridine ligand is redox non-innocent. The reaction eventually yields the oxido-bridged dichromium complex [K_4_(thf)_10_][Cr_2_*^t^*^Bu^L_2_(µ_2_-O)] (**40**), possibly after reactions with adventitious water and dihydrogen evolution as postulated in the previous reactions (Figure 12) [58]. The X-ray analysis of **40** revealed that the C–C bond distances around the 4-position of the pyridine ring (1.420(10)–1.439(10) Å) are much longer than the distances of the C2–C3 and C5–C6 bonds (1.352(9)–1.367(8) Å), indicating reduction in the pincer ligand *^t^*^Bu^L with an unpaired electron at the 4-position of the pyridine ring. The DFT calculation also supports the description of the oxidation state of **40** as Cr(II)(*^t^*^Bu^L^•−^) rather than Cr(I)*^t^*^Bu^L. 

In addition, reduction of **35** with two equiv of KC_8_ as well as oxidation of **35** with ferrocenium cation is examined [58]. Recrystallization of the reaction products again results in uptake of adventitious water to produce oxido-bridged tri- and tetranuclear complexes, respectively, with ambiguous reaction stoichiometry. The reduced species is trapped with carbon dioxide to afford a carbonato-bridged, dianionic–dinuclear complex [59]. On the other hand, the oxidation of **35** with a quinone gives rise to the formation of a bis(semiquinone) Cr(III)_2_ complex [58].

#### 2.1.2. Modified 1*H*-Pyrazol-3-yl Pincer Complexes

Partial replacement of the components in the protic pincer ligand ^R^LH_2_, the central pyridine and flanking pyrazoles, should tune the properties of their metal complexes as in other pincer-type complexes. This section provides an overview of the reactivities of protic pyrazole complexes obtained by ligand modification of ^R^LH_2_.

##### Modification at Pincer Center

Introduction of a strong σ-donor as the central ligating atom in the protic pincer framework should affect the reactivities of the trans ligand in particular. We reported ligand substitution of the NCN pincer-type ruthenium complexes **41**, which are obtained by cyclometalation of the corresponding 1,3-bis(1*H*-pyrazol-3-yl)benzenes (Figure 13) [46]. Owing to the trans effect of the central aryl group in the pincer ligand, the substitution takes place even at room temperature to give **42**, in contrast to the reaction of the NNN pincer-type analogue **4b** at 100 °C with the aid of a Lewis acid [60]. An iridium complex, bearing this NCN pincer ligand, is also synthesized in a similar manner [46].

We also installed an N-heterocyclic carbene (NHC) unit in the center of the pincer framework furnished with two protic pyrazoles [61]. The increase in the electron density of the resulting ruthenium complexes, such as **43**, is indicated by the CV measurements. Meanwhile, the CO stretching frequency of the carbonyl derivative is *higher* than that of the pyridine-centered counterpart [Ru(CO)(PPh_3_)_2_(*^t^*^Bu^L)] (1989 vs. 1964 cm^−1^). The DFT calculations suggest that the twist of the pincer ligand due to the increase in the chelate size in the NHC-centered pincer may lead to delocalization of the metal d-orbital to the pyrazole π-orbitals, which reduces π-back donation to the carbonyl ligand.

In addition to the C-centered pincer ligands, a novel triprotic NNN pincer ligand bearing a donating central nitrogen atom has been developed recently. The reaction of iron(II) chloride with 1,3-bis(1*H*-pyrazol-3-ylimino)isoindoline results in tautomerization of the ligand to give the 3-amino-1-imino-1*H*-isoindole complex **44** (Figure 14). This dichloridoiron(II) complex exhibits reactivities similar to those of the *^t^*^Bu^LH_2_ complex **19** shown in Figure 7 [62]. The CO stretching frequency of the carbonyl complex **46** (1963 cm^−1^) is much lower than that of the *^t^*^Bu^LH_2_ analogue **24a** (2005 cm^−1^) [49], indicating the donating nature of the central isoindole unit. Importantly, **46** undergoes deprotonation of the chelate backbone rather than the pyrazole groups to afford the isoindolin-2-yl complex **47** bearing a monoanionic bis(pyrazole) pincer ligand.

##### Unsymmetrical Pincer-Type Complexes

Replacement of one of the two pyrazole arms in pincer-type ^R^LH_2_ ligands by other donor groups has also been investigated. Such desymmetrization modifies the numbers and acidity of the protic sites as well as the electronic and steric properties of the metal center. We revealed that tautomerization of imidazole assisted by chelation with a pyrazolylpyridine unit results in the formation of the protic pincer-type ruthenium(II) complexes **48,** having protic pyrazole and N-heterocyclic carbene (pNHC [63]) arms (Figure 15) [60,64]. These complexes undergo reversible deprotonation to afford the corresponding pyrazolato complexes **49**, showing that protic pyrazole is more acidic than pNHC. Exhaustive deprotonation of **48a** under carbon monoxide gives the pyrazolato–imidazolyl carbonyl complex **50** [60]. Similar treatment of **48b** under dihydrogen results in heterolytic cleavage of H_2_ to yield the hydrido complex **51**, in which the proton derived from H_2_ goes to the more Brønsted basic imidazolyl arm [64].

A tertiary amino group can also be incorporated into the protic pincer framework to afford unsymmetrical pincer-type complexes with a hemilabile arm. After our isolation of the ruthenium(II) and iron(II) complexes, such as **52** [65], Goldberg and co-workers reported the reactivities of the platinum(II) complexes bearing this unsymmetrical protic pincer ligand (Figure 16) [47]. The methyl–pyrazolato complex **53**, obtained from a dimethylplatinum(II) complex and the free ligand, undergoes protonation at the pyrazolato arm rather than the methyl ligand, as observed in the symmetrical bis(pyrazolato) complex **16** (vide supra). The reaction of **54** with hydrogen chloride leads to methane evolution to afford the chlorido complex **55**. In contrast, intramolecular proton migration from the NH group to the methyl ligand appears more difficult. The methyl complex **54**, with a deuterium-labeled NH group, releases only unlabeled CH_4_ after being subjected to the temperature higher than 100 °C. The absence of CDH_3_ generation indicates that the methane is derived from an external proton source rather than the NH group. Interestingly, heating of the methyl–pyrazolato complex **53** in benzene results in the formation of the phenyl complex **56**. The reaction in C_6_D_6_ revealed concurrent site-specific deuteration of the methylene hydrogens in the ethyl groups of the pincer ligand, which implies the hemilabile nature of the dialkylaminomethyl arm.

Caulton and co-workers reported the iron [66] and cobalt [67] complexes bearing a PNN pincer-type protic pyrazole ligand (Figure 17). The dichlorido iron(II) and cobalt(II) complexes **57** with a distorted square-pyramidal geometry are deprotonated with lithium silylamide to afford the *N*-lithiated complexes **58**. Subsequent treatment with KC_8_ under carbon monoxide yields the iron(I) and cobalt(I) complexes **59**–**61**. Further reduction of the iron complex **59a** with KC_8_ results in the formation of a monoanionic dicarbonyl iron(0) complex. The iron complex **59a** also undergoes *N*-borylation of the pyrazolato arm with concurrent hydrogen evolution [66]. On the other hand, treatment of the mixture of the cobalt complexes **60** and **61** with an additional base leads to the second deprotonation at the methylene group of the pincer ligand to afford **62** with a dearomatized pyridine moiety [67], as in the related lutidine-based pincer-type complexes [17]. The result substantiates the diprotic nature of the PNN pincer-type ligand bearing a protic pyrazole arm.

The square-planar nickel(II) complex **63**, bearing a PNN pincer-type pyrazole ligand is also reported (Figure 18) [68]. As expected, deprotonation of **63** affords the corresponding pyrazolato complex **64**, which is subsequently converted to the azido complex **65**. Meanwhile, deprotonation of the SNN pincer-type complex **66** results in P–C bond cleavage of the pincer ligand to give the pyrazole complex **67** with a trigonal bipyramidal geometry. The methyl hydrogen atom as well as pyrazole proton is believed to be derived from adventitious water. The reaction in rigorously dried THF leads to redistribution of the nickel–pincer unit in **66** to give a bis(pincer ligand)-type nickel complex.

Previously, Lam and co-workers reported a CNN pincer-type pyrazole complex **68**, wherein one of the two protic pyrazolyl groups in the ^H^LH_2_ ligand is replaced by a phenyl group (Figure 19) [69]. This complex undergoes dehydrochlorination to afford the pyrazolato-bridged dinuclear complex **69**, as observed in bidentate 2-(1*H*-pyrazol-3-yl)pyridine complexes [70,71].

### 2.2. Redox Reactions of Hydrazines and Azobenzene

Given the expanded π-conjugated structure, the pincer-type ^R^LH_2_ ligands may be expected to function as an electron reservoir in addition to a proton source. The electron transfer coupled with proton transfer in the second coordination sphere appears crucial in various enzymatic transformations, including biological nitrogen fixation. In this context, reactions of protic pyrazole complexes with partially reduced dinitrogen species, such as hydrazine, have been investigated.

We revealed that the iron(II) *^t^*^Bu^LH_2_ complex **20a** catalyzes N–N bond cleavage of hydrazine as shown in Figure 20 [50]. The pivotal role of the pyrazole NH groups is suggested by the reactions of the N-methylated derivatives, **20a-Me** and **20a-Me2**, which are sluggish and much more complicated. The catalytic activities of the amide-substituted complex **70** [72] as well as the isoindoline-based pincer complex **45** [62] are also lower than that of **20a**. Control experiments and theoretical calculations [73] led to the proposed mechanism summarized in Figure 21, featuring multiple and bidirectional proton-coupled electron transfer (PCET) between the metal–ligand bifunctional platform and the hydrazine substrate. The pyrazole NH group promotes heterolytic N–N bond cleavage of the coordinated hydrazine in **71** through a hydrogen bond with the distal nitrogen atom. The second pyrazole NH group in the pincer ligand behaves as an acid–base catalyst for substitution of the amido ligand in **72** by the second molecule of hydrazine to afford the hydrazido(1−) complex **73**. The calculations also suggest some radical character of the κ*N*-nitrogen ligands in these high-valent iron species **72** and **73**, and hence a mixed electronic structure of Fe^IV^(NH_2_^−^) and Fe^III^(NH_2_^•^), for example [73]. Following PCET from the hydrazido(1−) ligand to the high-valent iron–bis(pyrazolato) fragment would yield the iron(II) diazene complexes **74**. In fact, the phenyldiazene complex **74b** is isolated in the reaction of phenylhydrazine. An X-ray analysis revealed that the phenyldiazene ligand in **74b** benefits from stabilization by hydrogen-bonding interactions with the two pyrazole NH units and counteranion. Meanwhile, the reaction of 1,1-diphenylhydrazine results in reductive elimination of a hydrazinophosphonium salt from the hydrazido(1−) iron(IV) complex **73** bearing trimethylphosphine ligands (omitted in Figure 21) due to the lack of the distal hydrogen atom. Finally, the diazene complex **74a** releases free diazene, which disproportionates to dinitrogen and hydrazine. An alternative scenario that merits comments involves a direct reaction of **74a** with hydrazine to give two moles of ammonia. The proposed 2H^+^/2e^−^ shuttling may be applicable to other multielectron redox processes.

Similar disproportionation of a substituted hydrazine is reported for a triprotic, tripodal tris(1*H*-pyrazol-3-ylmethyl)amine complex [74]. Treatment of the chlorido-bridged diruthenium(II) complex **75** with 1,2-diphenylhydrazine results in N–N bond cleavage to afford the aniline complex **76** (Figure 22) [75]. Concurrent formation of azobenzene along with free aniline indicates disproportionation of two moles of 1,2-diphenylhydrazine to azobenzene and two moles of aniline in this transformation. The aniline ligand in **76** is engaged in hydrogen bonding network with the protic pyrazole units along with the chloride counteranion. The disproportionation proceeds catalytically when an excess of 1,2-diphenylhydrazine is added to **75**. An analogous complex of a tetradentate ligand having non-protic pyridylmethyl arms displays no catalytic activity, even in the presence of external protic pyrazole as a proton source, indicating that the proton-responsive unit in the second coordination sphere is responsible for the N–N bond cleavage.

Caulton’s group demonstrated that the reaction of dichromium(II) complex **35** (vide supra) [57] with azobenzene yields the paramagnetic chromium(III) complex **77** having an unsymmetrically bridged PhNNPh unit (Figure 23) [76]. The N–N distance of 1.471(9) Å indicates that the N=N bond is reduced by the chromium(II) centers in **35** to give the hydrazido(2−) ligand. A benzo[*c*]cinnoline derivative, featuring η^2^:η^2^-coordination of the azo group and lack of the THF ligand, is also characterized. Reduction of these hydrazido(2−) complexes with KC_8_ is further examined. Only the benzo[*c*]cinnoline derivative undergoes N–N bond cleavage upon treatment with an excess of the reductant.

Electron transfer from the dichloromium(II) complex **35** to phenylhydrazine gives rise to the formation of the phenylhydrazido(2−)-bridged di(aniline)dichromium(III) complex **78** (Figure 23) [77]. The two aniline ligands, derived from N–N bond cleavage of phenylhydrazine, bind to one of the two chromium atoms at trans positions and form intramolecular hydrogen bonding with the protic pyrazoles on the other chromium center. While the reaction stoichiometry should be rather complicated, ammonia and benzene were detected as the fission products in the reaction mixture.

Even dinitrogen is coordinated to proton-responsive pyrazole-based complexes, as in the mononuclear *^t^*^Bu^L complex **8** [40]. The NCN pincer-type dinuclear complex **79**, supported by two linking diphosphine ligands, reacts with dinitrogen to afford the dinitrogen-bridged diruthenium(II) complex **80** (Figure 24) [78]. Unfortunately, no further transformation of the dinitrogen ligand in this multiproton-responsive cavity has not been reported.

### 2.3. Nitrate Reduction

Reduction of nitrogen oxides constitutes the inorganic nitrogen cycle in nature. Better understanding of the reaction may be important to address the issue of the coastal eutrophication caused by excessive use of nitrogen fertilizer. We found that the reaction of the ruthenium(II) complex **81** with silver nitrite leads to N–O bond cleavage of the nitro anion, giving the nitrosylruthenium(II) complex **82** (Figure 25) [41]. The dehydrative conversion is most likely assisted by proton transfer from the pyrazole ligand, and the overall reaction is redox neutral. 

Caulton’s group demonstrated that *N*,*N*′-disilyldihydropyrazines, whose usefulness in salt-free reduction was uncovered by Mashima and co-workers [79], are effective for deoxygenation of NO*_x_* ligands on protic pyrazole complexes. Thus, the tris(nitrate) chromium(III) complex **83,** bearing a protic bis(pyrazole)-type pincer ligand [80], reacted with three equiv of an *N*,*N*′-disilyldihydropyrazine to afford the nitrosyl–nitrate chromium(I) complex **84** (Figure 26) [81]. The reaction byproducts, hexamethyldisiloxane and pyrazine, were detected in the reaction mixture via ^1^H NMR spectroscopy. It is to be noted that the reaction takes place without any damage on the pyrazole NH protons.

Similarly, treatment of the bis(1*H*-pyrazol-3-ylpyridine)nickel(II) nitrato complex **85** with an equimolar amount of the disilyldihydropyrazine results in deoxygenation of the nitrate and formation of mono-deprotonated product **86** (Figure 27) [71]. The NH proton remaining on a pyrazole ring in **86** is identified by the larger N_α_Ν_β_C angle. The deoxygenated product in this transformation, nitrite ion, was not detected; however, further deoxygenation of **86** affords the diamagnetic nickel(0) complex **87**, having linear nitrosyl ligands derived from deoxygenation of the nitrate ligand. Thermodynamics of deoxygenation of the NO*_x_* ligands with a disilyldihydropyrazine was also investigated theoretically for manganese complexes bearing a PNN pincer-type protic pyrazole ligand [82].

The PNN pincer-type cobalt(II) dichlorido complex **88** bearing a protic pyrazole arm reacts with sodium nitrite to afford the tris(nitrito-*N*)cobalt(III) complex **89** and bent nitrosyl cobalt(III) complex **90** [83]. Although the yields and ratio of **89** and **90** are not described, the reaction stoichiometry shown in Figure 28 has been proposed along with proton transfer from the pyrazole moiety to the bridging nitrite in a dinuclear intermediate. The compounds **89** and **90** are independently obtained by the reaction of the pincer ligand with a cobalt(III) complex Na_3_[Co(NO_2_)_6_] followed by deoxygenation and deprotonation with a disilyldihydropyrazine [83].

### 2.4. CO_2_ Reduction

Liaw, Lu, and co-workers demonstrated that the pyrazolato-bridged {Fe(NO)_2_}^9^ complex **91** undergoes two-electron reduction to give the dianionic complex **92** (Figure 29) [84]. Subsequent reaction with carbon dioxide results in nucleophilic attack of the pyrazolato ligand, resulting in the formation of the mononuclear CO_2_ adduct **93**. Interestingly, addition of an equimolar amount of calcium triflate yields calcium oxalate and regenerates the dinuclear complex **91**. A calcium-assisted one-electron transfer from the iron center to the CO_2_ unit followed by bimolecular coupling with **93** is proposed for the last step of this synthetic cycle.

## 3. Catalysis of Protic Pyrazole Complexes

In addition to the aerobic oxidation of phosphines (Section 2.1.1) and catalytic disproportionation of hydrazines (Section 2.2), the catalytic application of protic pyrazole complexes to various chemical transformations has been investigated. This section focuses on recent work along with studies in which the role of the protic pyrazole ligand is evident or discussed. A previous review on this topic is available [11].

### 3.1. Hydrogenation and Transfer Hydrogenation

In 2008, Yu’s group demonstrated that the protic pincer-type ruthenium(II) complex **94** catalyzes transfer hydrogenation of acetophenones with 2-propanol in the presence of an excess of a base (Figure 30) [85]. Introduction of a non-protic pyrazole arm instead of the NHC in **94** significantly accelerates the reaction with **95** [86,87]. It is to be noted that the imidazole complex **96a** with an NH group at a remoter position γ to the metal displays catalytic activity comparable with **95**, whereas the non-protic analogue **96b** is much less effective [88,89]. These results may suggest an inner-sphere mechanism involving β-hydrogen elimination of an alkoxide intermediate instead of a pyrazole-aided outer-sphere hydrogen transfer. The NH group in the pincer ligand may still operate to facilitate the dissociation of the halido ligand and to increase the nucleophilic character of the hydrido intermediate through deprotonation of the NH unit. Even asymmetric transfer hydrogenation of aryl ketones has been achieved with protic pincer-type complexes, such as **97**, bearing an optically active oxazolinyl group in the chelate framework (Figure 31) [90]. We [46] and Halcrow [91] also reported that ^R^LH_2_ and related NCN pincer-type ruthenium(II) complexes **4** and **41** promote catalytic transfer hydrogenation of acetophenone with 2-propanol in the presence of alkoxide bases. The reactivities parallel with stoichiometric hydrogenation transfer in other protic pyrazole complexes [92,93].

Thiel and co-workers demonstrated that the diprotic ruthenium(II) complexes **4a** and **98** catalyze not only transfer hydrogenation but also hydrogenation of acetophenone (Figure 32) [39]. Theoretical calculations suggest that outer sphere hydrogen transfer from the anticipated hydrido–pyrazole intermediate to the carbonyl substrate as well as the heterolytic cleavage of dihydrogen at the coordinatively unsaturated pyrazolato complex is facile. On the other hand, in the transfer hydrogenation reaction, the efficiency of the non-protic analogues, such as **99**, is comparable or even higher in some cases [94,95,96,97], implying that such metal–pyrazole cooperating mechanism is less probable. Catalytic hydrogenation with the *^t^*^Bu^LH_2_ analogue **4b** is also reported [65].

Nikonov’s group uncovered that the ruthenium(II) complexes, typified by **100**, bearing a P–N chelate protic pyrazole ligand, promote transfer hydrogenation of not only acetophenone [98] but also nitriles, heteroaromatics, alkynes, alkenes, and esters [99]. In the reaction of aromatic and aliphatic nitriles with 2-propanol, the initially formed amine product further reacts with acetone, a byproduct of the transfer hydrogenation in 2-propanol, to afford the ketimines as the final product (Figure 33). No further reduction of the ketimines to secondary amines is observed. In the transfer hydrogenation of inner alkynes, semi-hydrogenation products, alkenes, are obtained with *E*-selectivity. The *E*-alkenes would be formed by the isomerization of *Z*-alkenes initially generated. Actually, *cis*-stilbene isomerizes to *trans*-stilbene under the catalytic conditions. When the substituents on alkynes are less bulky, further reduction takes place to give the corresponding alkanes. Meanwhile, the conversion of a terminal alkyne is very low. Figure 33 also illustrates catalytic transfer hydrogenation of ethyl trifluoroacetate to 1,1,1-trifluoroethanol. In this reaction, ethanol is oxidized to ethyl acetate through Tishchenko reaction, and whole process can be viewed as ester metathesis [100].

Gong, Meggers, and co-workers revealed that the chiral-at-metal iridium(III) complex **101** catalyzes asymmetric transfer hydrogenation of ketones in the presence of protic pyrazoles (Figure 34) [101]. Other additives, such as P*n*Bu_3_, 2,6-diaminopyridine, and imidazole, leads to much lower catalytic activity and enantioselectivity as the reaction without the protic pyrazoles. An X-ray analysis of a related chlorido–pyrazole complex suggests that the NH unit would be placed in the optimum orientation for efficient bifunctional hydrogen transfer to the ketone substrate (inset). Further, attractive π–π interaction between the C–N chelate ligand and the arene ring in the substrate is proposed to realize the high enantioselectivity. This binary catalyst system is also effective for asymmetric hydrogenation of acetophenone [102]. In addition, the catalyst serves as a photoredox mediator, which allows asymmetric hydrogenation and photoredox transformation sequences without isolation of the chiral alcohol intermediate. On the other hand, half-sandwich C–N chelate pyrazole complexes of iridium(III) are known to catalyze transfer hydrogenation of acetophenone with 2-propanol [93].

Niedner-Schatteburg, van Wüllen, and Thiel’s team demonstrated that the pyrazolatoruthenium(II) complex **102** catalyzes hydrogenation of carbon dioxide under supercritical conditions (Figure 35), in addition to transfer hydrogenation of acetophenone with 2-propanol [103]. The CO_2_ hydrogenation activity of **102** is almost comparable with those of the conventional dichloridoruthenium(II) phosphine complexes. A mechanism without any proton response of the pyrazolato group is proposed on the basis of theoretical calculations (Figure 36). A major role of the ionizable pyrazole in the chelate appears to be to make the chelate ligand anionic and more electron-donating [104]. The stronger trans influence of the pyrazolato moiety leads to generation of a vacant site, where carbon dioxide is coordinated. Subsequent insertion into the Ru–H bond gives a formato ligand, which mediates heterolytic cleavage of dihydrogen.

Himeda, Ertem, and co-workers described the half-sandwich iridium(III) complexes bearing proton-responsive ligands as efficient CO_2_ hydrogenation catalysts in basic aqueous solutions (Figure 37) [105,106]. The protic pyrazole group brings about better catalytic activity than the *N*-methylpyrazole group (**103a** vs. **103b** and **104a** vs. **104b**). The performance of the pyrazole complexes **103** is, however, not so prominent when compared with that of the imidazole complexes **105**. Meanwhile, introduction of an OH group at the 6-position of the pyridine ring significantly accelerates the reaction (**104** and **106**). These observations as well as DFT calculations led to the mechanistic proposal that the proton-responsive site on the diazole rings offers strong electron donation to the iridium center through its deprotonation. Still, the diazolato unit may also provide a proton acceptor site for the H_2_ heterolysis even in the absence of an OH group.

### 3.2. Hydrogen Evolution

Dehydrogenation of formic acid, a reverse reaction of CO_2_ hydrogenation, is also catalyzed with protic pyrazole complexes. Wang, Himeda, and co-workers revealed that the 2-(1*H*-pyrazol-3-yl)pyridine iridium(III) complex **107** promotes hydrogen evolution from formic acid under acidic conditions (Figure 38) [107]. The catalytic activity of **107** is less than that of the imidazole analogue **108** with a remoter γ-NH group, suggesting that the azole units mainly have a role in increasing the electron donation to the metal center through their deprotonation. Interestingly, introduction of a pendant pyridyl group on the pyrazole unit improves the catalytic activity [108]. The proposed mechanism involves a two-point hydrogen bonding between protic pyrazole–pyridinium unit and an external formic acid in the second coordination sphere (Figure 39). The proton relay would facilitate the protonation to the hydrido ligand.

We reported hydrogen evolution from formic acid catalyzed by the ^CF3^LH ruthenium(II) complex **109** (Figure 40) [109]. The catalytic activity of the less acidic *^t^*^Bu^LH analogue **6** is poor, indicating the importance of proton transfer from the protic pyrazole arm in the catalysis. The reaction with the NCN pincer-type complex **110** is also much slower.

In addition to formic acid, amine–boranes have attracted much attention in terms of chemical H_2_-storage. Pal and Nozaki reported hydrogen evolution from dimethylamine–borane promoted by the pyrazole–pyrazolato rhodium(III) complex **111** [110]. On the basis of theoretical calculations as well as the fact that [{Cp*RhCl_2_}_2_] and the free pyrazole are catalytically inactive in separate runs, a metal–pyrazole cooperative mechanism is proposed, as shown in Figure 41. Dehydrochlorination of the catalyst precursor **111** generates the coordinatively unsaturated bis(pyrazole) complex **112**. The Lewis acidic rhodium center and Brønsted basic pyrazolato ligand in **112** dehydrogenates dimethylamine–borane substrate in a cooperative manner. The resulting hydrido complex **113** releases dihydrogen gas guided by an intramolecular hydrogen bond to regenerate **112**.

### 3.3. Borrowing Hydrogen Catalysis

Owing to their hydrogen transfer ability, protic pyrazole complexes also exhibit borrowing hydrogen catalysis, wherein the catalyst first borrows hydrogen atoms from the alcohol substrate and then returns them after the bond formation of the resulting carbonyl intermediate [111]. Ryu and co-workers applied a pincer-type ligand in the Yu’s transfer hydrogenation catalyst (Section 3.1) to α-alkylation of amides with primary alcohols (Figure 42) [112]. As in typical borrowing hydrogen transformations, initial dehydrogenative oxidation of the primary alcohol followed by dehydrative condensation of the resulting aldehyde and amide is proposed. Subsequent transfer hydrogenation from the catalyst would yield the α-alkylation product.

Bagh’s group demonstrated that the pyrazolato-bridged diiridium(III) complex **114** promotes α-alkylation of arylacetonitriles with secondary alcohols (Figure 43) [113]. When the catalyst precursor **114** is dissolved in DMSO, the DMSO complex **115** is obtained, whereas reaction of **114** with cyclohexanol affords the hydrido–pyrazole complex **116** with liberation of cyclohexanone (Figure 43b). Formation of these mononuclear N–O chelate complexes suggests that the catalysis involves initial split of the pyrazolato dimer complex **114** into a coordinatively unsaturated pyrazolato complex **117**, which then undergoes transfer hydrogenation from the secondary alcohol to give the hydrido–pyrazole complex **116** (Figure 43c). Following steps in borrowing hydrogen cycle were also supported by the stoichiometric reactions.

### 3.4. Dehydrogenative Oxidation

If the hydride intermediate in the borrowing hydrogen catalysis somehow releases dihydrogen gas instead of returning the hydrogen atoms to give the redox-neutral product, the net reaction would turn to be dehydrogenative oxidation. Hölscher, Bera, and co-workers described acceptorless double dehydrogenation of primary amines catalyzed by the ruthenium(II) complex **118a** bearing an N–N chelate protic pyrazole ligand (Figure 44) [114]. Both aromatic and aliphatic nitriles are obtained in this manner. Secondary amines are also converted to the corresponding imines under the reaction conditions. The poor activity of the *N*-methylated analogue **118b** indicates the crucial role of the pyrazole NH group in the catalysis. Meanwhile, the uncharged pyrazolato complex **119** with a Lewis acid displays catalytic performance comparable with that of **118a** even in the absence of base. A coordinatively unsaturated pyrazolato complex is thus suggested to be a catalytically active species. Computational study proposed that the pyrazole unit is not involved directly in the abstraction of hydride from the amine substrate, but dehydrogenation of an imine intermediate occurs in a concerted, metal–ligand cooperative manner with the aid of an external substrate molecule in the second coordination sphere (inset). The resultant hydrido complex would undergo protonation with ammonium cation to evolve dihydrogen gas.

Chai and co-workers reported synthesis of imines catalyzed by the protic pyrazole manganese(I) complex **120a** (Figure 45) [115]. The non-protic analogue **120b** exhibits lower conversion and brings about increased formation of the amine byproduct through the borrowing hydrogen pathway, although the detailed reason is not mentioned. The reaction is proposed to proceed via dehydrative condensation of aniline and benzaldehyde, which is formed by dehydrogenative oxidation of benzyl alcohol with a coordinatively unsaturated pyrazolato complex.

### 3.5. Transformation of Allylic and Propargylic Compounds

Satake and co-workers demonstrated that the N–N chelate pyridylpyrazole palladium(II) complex **121** catalyzes the reaction of allylic acetates and ketene silyl acetals, as shown in Figure 46 [116]. The significance of the NH group therein is evident by comparison with the catalytic performance of the non-protic complex **122**. On the other hand, the position of the NH group appears unimportant since the protic imidazole complex **123** bearing a γ-NH group displays similar catalytic activity and even better selectivity for cyclopropanation over allylation [117]. The proposed mechanism is illustrated in Figure 47. Deprotonation of the catalyst precursor **121** gives the uncharged pyrazolato complex **124**, which undergoes nucleophilic attack of the ketene silyl acetal to afford the palladacyclobutane complex **125**. The major role of the NH group would be to make the chelate ligand a better σ-donor, which directs the attack of nucleophile at the central carbon rather than the terminal carbon atoms [118]. Use of chiral oxazolidines as the chelate tether realizes asymmetric cyclopropanation of a ketene silyl acetal with moderate stereoselectivity [119].

Gimeno, Lledós, and co-workers described isomerization of allylic alcohols mediated by protic azole ruthenium(IV) catalyst precursors in water (Figure 48) [120,121]. Because the *N*-methylated pyrazole and γ-protic imidazole complexes exhibit similar and much higher catalytic performance, respectively, the aqua ligand rather than the protic azoles is proposed as an acid–base catalytic site to promote the metal–ligand cooperative hydrido migration from the allylic carbon to the vinyl carbon atom.

We uncovered that the protic pyrazole ruthenium(II) complexes **126** catalyze isomerization of 1,1-dimethyl-3-phenylprop-2-yn-1-ol to 3-methyl-1-phenylbut-2-en-1-one in methanol (Figure 49) [122]. The reaction, known as Meyer–Schuster rearrangement [123,124], does not occur with non-protic analogue **127** as well as a substitution-inert isocyanide complex, whereas the complex **126b** with a less electron-withdrawing phenyl substituent requires a more elevated temperature (reflux, 87%). These observations indicate that both Lewis acidic metal center and Brønsted acidic NH group in **126** are necessary for this catalysis. A proposed mechanism is shown in Figure 50. The π-bound propargylic alcohol in **128** undergoes an S_N_2′-type propargylic substitution by the solvent methanol with the aid of the intramolecular NH···O hydrogen bonding in the second coordination sphere to afford the allene complex **129**. Dissociation and solvolysis of the allene would yield an acetal, which is finally converted to the enone product via hydrolysis.

Use of less nucleophilic solvents, such as 1,2-dichloroethane and 1,4-dioxane, completely switches the reaction outcome. In the absence of the external nucleophile, a two-point interaction between the protic pyrazole complex and the substrate would lead to facile C–O bond cleavage, giving the η^3^-propargyl complex **130** (Figure 51). Involvement of **130** in the Meyer–Schuster rearrangement in methanol (Figure 50) is less likely because the nucleophilic addition to η^3^-propargyl ligands generally takes place at the central carbon atom [125] and would fail to provide the observed product. The pyrazolato unit in **130** mediates proton migration in the η^3^-propargyl ligand to afford the η^3^-butadienyl complex **131**, which is isolable in the case of R = CF_3_. When the pyrazolato ligand in **131** is more nucleophilic (R = Ph), intramolecular addition to the terminal carbon in the η^3^-butadienyl ligand occurs to give the *N*-allenylmethylpyrazole complex **132**.

### 3.6. Hydroamination of Alkenes

The metal–ligand bifunctional nature of protic pyrazole complexes is further applied to catalytic hydroamination of alkenes, in which both amine and olefin functional groups are activated simultaneously. We disclosed that the C–N chelate protic pyrazole iridium(III) complex **133** promotes cyclization of aminoalkenes with the aid of an equimolar amount of an alkoxide base (Figure 52). The catalyst is compatible with various functional groups such as ester, bromo, cyano, and hydroxy groups. The pyrazolato-bridged dimer **134**, obtained by dehydrochlorination of **133**, exhibits catalytic activity similar to **133** even in the absence of the base. Meanwhile, the catalytic performance of the related complexes, **135** and **136,** having a proton-responsive site at the positions γ and α to the iridium center, respectively, is poor. Additionally, catalytically inactive are the *N*-methylated derivative **137** and the six-membered chelate analogue **138** [93]. These results indicate that an exquisitely positioned proton-responsive site with appropriate direction and acidity is crucial for the catalysis. Figure 53 illustrates a proposed mechanism featuring the metal–ligand cooperation. The olefin part in the aminoalkene substrate binds to the coordinatively unsaturated mononuclear pyrazolato complex **139**, derived from dehydrochlorination of the chlorido complex **133** or split of the pyrazolato dimer **134**. The activated olefin is attacked by the amino group with increased nucleophilic character owing to intramolecular hydrogen bond with the pyrazolato unit. The tight-fitting assembly in transition state **140** is supported by the large negative activation entropy provided by kinetic experiments. Subsequent proton transfer from the pyrazole unit to the Ir–C bond yields the product and regenerates the unsaturated pyrazolato complex **139**.

### 3.7. Hydration of Nitriles

Metal-mediated coupling of nitrile and pyrazole to give a stable chelate pyrazolylamidino ligand, illustrated in Figure 54, has been known for a long time. Nevertheless, a certain type of protic pyrazole complex catalyzes hydration of nitriles. Rodríguez, Romero, and co-workers reported hydration of benzonitriles and acrylonitrile catalyzed by the pyrazole complexes, such as **141** and **142** (Figure 55) [126,127]. The reaction is proposed to proceed via nucleophilic attack of water or hydroxide anion to the coordinated nitrile. Although the role of the pyrazole ligand in this catalytic reaction is not mentioned, the proton-responsive pyrazole ligand may increase the nucleophilic character of water through hydrogen bonding in the second coordination sphere.

### 3.8. Catalysis with Coordinatively Saturated Complexes

In their seminal work on chiral-at-metal complexes [128,129], Gong and Meggers described ligand-centered catalysis of protic pyrazole complexes. For example, the iridium complex **143** bearing an amido-substituted protic pyrazole ligand catalyzes asymmetric 1,4-addition of indoles to β-nitroacrylates (Figure 56) [130]. As the well-known thiourea organocatalysis [131], the amidopyrazole unit serves as a hydrogen bond donor to the nitroalkene. The chiral metal center in the catalyst **143** is coordinatively saturated and inert; however, the carboxamide substituent in the C–N chelate acts as a hydrogen bond acceptor for the indole, making the complex bifunctional.

### 3.9. Miscellaneous

Suzuki–Miyaura coupling reactions with bis(pyrazole) palladium(II) complexes [132] as well as a binary system of PdCl_2_ and ^Ph^LH_2_ [133] are reported. Garralda and co-workers described hydrolysis of ammonia–borane and amine–boranes catalyzed by the rhodium(III) bis(pyrazole) complexes, such as **144** (Figure 57) [134]. Rodríguez and Romero reported the ruthenium(II) complex **145** bearing a protic pyrazole ligand catalyzes photochemical oxidation of alcohols in water in the presence of a sacrificial oxidant along with [Ru(bpy)_3_]^2+^ as a photosensitizer (Figure 58) [135]. Chemical oxidation of alkenes to epoxides is also promoted. Unfortunately, the mechanisms for these reactions and the role of the protic pyrazole ligand therein are not discussed.

## 4. Conclusions

It is now apparent that the protic pyrazoles are versatile non-innocent ligands, owing to their ability to place a proton-responsive site in the second coordination sphere. The deprotonation of the β-NH unit turns the pyrazole ligand to a stronger σ-donor. In both protonated and deprotonated forms, the ligand-based hydrogen bonding and electrostatic interactions at the β-position allows efficient substrate recognition and activation as well as additional tuning of the electronic properties of the metal center. Importantly, these events in the outer coordination sphere are linked with the metal-centered reactions in some cases. Such metal–ligand cooperation can be compared with those of related proton-responsive ligand systems (Figure 1); however, the stiff five-membered structure that fixes the proton with an azole acidity in the metal–pyrazole plane makes the reactivity of protic pyrazole complexes distinctive. Additionally, facile redox of the metal–pyrazole framework coupled with deprotonation is implied for π-delocalized, bis(pyrazole)-type *^t^*^Bu^LH_2_ complexes (Figure 21 and compound **40** in Figure 12). The metal–ligand cooperative catalysis thus has been explored for various types of reactions, typified by transfer hydrogenation and functionalization of unsaturated carbon–carbon bonds. The role of the NH groups therein, however, appears to depend on the catalyst systems. In some reactions, control experiments with γ-protic imidazole complexes indicate that the presence of the ionizable NH group is prerequisite for the catalysis but the position is unimportant. The protic pyrazole ligand in this case would remain deprotonated during the catalytic turnover and serve only to increase the electron density of the metal center. Further studies are needed to elucidate the conditions under which the metal–ligand cooperation is exerted. These efforts will lead to improvement and deeper mechanistic understanding of the catalysis of protic pyrazole complexes. The metal–ligand cooperation has also been applied to activation of small inorganic molecules. A future target in this direction will undoubtedly be multiproton-coupled multielectron reduction of inert molecules, such as carbon dioxide and dinitrogen, with polyprotic pyrazole complexes. In these studies, novel design of protic pyrazole ligands will continue to be a major subject to create more sophisticated metal–ligand cooperation platforms for both stoichiometric and catalytic transformations unique to protic pyrazole complexes. Facile construction of the pyrazole rings will be advantageous in the synthesis of the pyrazole ligands whose Brønsted acidity and deployment are controlled by the substituents on the ring and chelate framework.

## Data Availability

Data sharing is not applicable to this article.

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
