# Peer review of "Recent Developments in Reactions and Catalysis of Protic Pyrazole Complexes"

_molecules, 2023, doi:10.3390/molecules28083529_

Round 1
Reviewer 1 Report
1. The review is organized into different parts, from the synthesis approaches to the different potential applications. A general discussion about the main advantages and current limitations on this Protic pyrazoles complexes. 2. There are many figures; I suggest the author trim them. 3.“Coordination of a pyrazole to a Lewis acidic metal center renders the pyrazole NH proton more acidic.” This illustration should be cited the refs, such as Molecules, 2019, 24, 1760; and Dyes and Pigments, 2016, 134, 139-147 and Chem. Commun., 2022, 58, 6653–6656 4. Conclusion and challenges section seems to be simple. Pls do it in detail 5. At the end of the review, the author summarizes the main progress, but perspectives for the future development are missing. 6. There are some grammatical and writing errors in the manuscript.
Reviewer 2 Report
This review is devoted to the chemistry of complexes of late transition metals with pyrazole ligands containing an unsubstituted nitrogen atom in the pyrazole ring. These complexes are very important in modern organic chemistry, as they are effective catalysts for hydrogen transfer reactions. The authors quite correctly consider precisely β-protonated pyrazole complexes, since their properties differ markedly from those of complexes with N-substituted pyrazole fragments. I believe that such a review will be in demand by chemists working both in the field of coordination chemistry and in the field of organic chemistry.
I have a few comments:
1. Scheme 21. When compound 73 from compound 72 is formed, it is necessary to show where Me3P comes from. The catalytic complex itself cannot serve as its source, since it is catalysis and its quantity is insufficient to obtain stoichiometric quantities of the product. Also, the principle of electrical neutrality and the material balance for R and R2NNH2 are not observed.
2. Scheme 22. The material balance for groups PhNH is not observed.
3. Scheme 43C. Typo: Drawn acetone instead of isopropanol at the top of the cycle.
4. Section 3.8. I believe that iridium complex catalysis should not be called "organocatalysis" even if it was called that in the original article. If there were no metal center, there would probably be no catalysis. And, in any case, there would be no asymmetric induction. In my opinion, the term "organocatalysis" in this case is erroneous.
In general, I can recommend the manuscript for publication in Molecules, after taking into account the comments made.
Reviewer 3 Report
This review focuses on the reactivity of metal complexes with pyrazole-containing chelating ligands involved in proton transfer. As such, this review provides design guidelines for catalysts that can achieve efficient proton-conjugated reactions and will be of great interest to many chemists working in the fields of coordination chemistry, organometallic chemistry, and molecular catalysis chemistry, and is worthy of publication in Molecules. The first part on complex chemistry reactivity is organized according to the structure of the ligand and the central metal, and the second part on catalysis is discussed by reaction type, making the book very easy to read. Therefore, no special modifications are needed.
Author Response
Point 1: ... Therefore, no special modifications are needed.
Response 1: We appreciate positive consideration of the reviewer.